# DOWNSTREAM DATASETS MAKE SURPRISINGLY GOOD PRETRAINING CORPORA

## ABSTRACT

For most natural language processing tasks, the dominant practice is to fine-tune large pretrained transformer models (e.g., BERT) using smaller downstream datasets. Despite the success of this approach, it remains unclear to what extent these gains are attributable to the massive background corpora employed for pretraining versus to the pretraining objectives themselves. This paper introduces a large-scale study of *self-pretraining*, where the same (downstream) training data is used for both pretraining and finetuning. In experiments addressing both ELECTRA and RoBERTa models and 10 distinct downstream classification datasets, we observe that self-pretraining rivals standard pretraining on the BookWiki corpus (despite using around $10\times$–$500\times$ less data), outperforming the latter on 7 and 5 datasets, respectively. Surprisingly, these task-specific pretrained models often perform well on other tasks, including the GLUE benchmark. Besides classification tasks, self-pretraining also provides benefits on structured output prediction tasks such as span based question answering and commonsense inference, often providing more than 50% of the performance boosts provided by pretraining on the BookWiki corpus. Our results hint that in many scenarios, performance gains attributable to pretraining are driven primarily by the pretraining objective itself and are not always attributable to the use of external pretraining data in massive amounts. These findings are especially relevant in light of concerns about intellectual property and offensive content in web-scale pretraining data.

## 1 INTRODUCTION

For training predictive models operating on natural language data, the current best practice is to *pretrain* models on large unlabeled *upstream* corpora to optimize self-supervised objectives, for example, masked language modeling (MLM); the resulting weights are then used to initialize models that are subsequently trained (*finetuned*) on the labeled *downstream* data available for the task at hand. Large-scale pretrained models typically provide significant performance boosts when compared to models trained directly on the downstream task (with random initializations) (Peters et al., 2018; Devlin et al., 2019; Chiang & Lee, 2020; Krishna et al., 2021). Upstream corpora tend to be significantly larger than the downstream corpora and the success of this approach is often attributed to its ability to leverage these massive upstream corpora (Liu et al., 2019; Yang et al., 2019). For example, the seminal BERT model (Devlin et al., 2019) was pretrained using the BookWiki corpus which is a combination of English Wikipedia and BooksCorpus (Zhu et al., 2015), totaling 13GB of plain text. Subsequent models have moved on to web-scale data. For example, XLNet (Yang et al., 2019), RoBERTa (Liu et al., 2019), and T5 (Raffel et al., 2020)), were trained on 158GB, 160GB and 750GB of data, respectively.

As upstream corpus size and downstream performance have gone up, popular attempts at explaining these gains have focused on themes of "knowledge transfer" from the upstream corpus, attributing them to shared linguistic structure, semantics (Lina et al., 2019; Tenney et al., 2019), and facts about the world (Petroni et al., 2019). However, since the introduction of large-scale pretraining corpora occurred together with the invention of self-supervised pretraining objectives (e.g. masked language modeling (Devlin et al., 2019) and replaced token detection (Clark et al., 2019)), it remains unclear to what extent large-scale corpora are integral to these leaps in performance. For several tasks, especially summarization, recent works have managed to achieve surprising performance gains in settings where

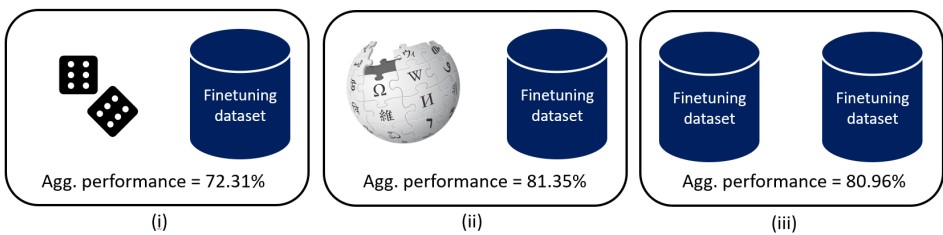

Figure 1: Aggregate performance of an ELECTRA model across 10 finetuning datasets when it is (i) randomly initialized (ii) pretrained on upstream corpus (BookWiki) (iii) pretrained on the finetuning dataset itself

the upstream corpus is created synthetically with arbitrary symbols, but the pretraining objective is designed to capture some of the structure of the task (Krishna et al., 2021; Wu et al., 2022).

In this work, we ask just how much of pretraining's benefits could be realized in the absence of upstream corpora by pretraining directly on the downstream corpora (with the same self-supervised objectives). We find that this approach, which we call *self-pretraining*, often rivals the performance boosts conferred by *off-the-shelf* models pretrained on large upstream corpora (Figure 1), even outperforming them on 7 out of 10 datasets. Prior research has shown that *additional* self-supervised pretraining of off-the-shelf models using the downstream data can give further gains (Gururangan et al., 2020). Yao et al. (2022) showed that one can use the downstream data to retrieve a tiny subset of a large general corpus for pretraining effciently without sacrificing performance. Our study goes further, showing that even when starting from random initializations, and without using any external data beyond the downstream data itself, self-pretraining can rival standard practices. Since self-pretraining requires the same data that must already be available for downstream finetuning, the benefits of pretraining in this case cannot be attributed to *transfer* of knowledge from the upstream corpus. Instead, these benefits can only be attributed to the pretraining objective, which is possibly able to learn some inductive biases better than the finetuning objective (e.g. linguistic knowledge Tenney et al. (2019)), or perhaps simply initialize network parameters such that their statistics lead to better optimization during finetuning (Wu et al., 2022). While we note that similar observations have been made in the computer vision community (El-Nouby et al., 2021), we argue that it is especially important to establish these phenomena in the language domain, for which building on self-supervised pretrained models is now the ubiquitous practice of the vast majority of practitioners.

To understand differences in predictions with different pretraining strategies (i.e., between self-pretrained and off-the-shelf models), we analyse the errors made by these models on the same downstream data. Despite similar performance of these models, we find that self-pretrained and off-the-shelf models make significantly less correlated errors when compared to two independently finetuned models pretrained with either strategy. However, we observe that these uncorrelated mistakes do not transfer to improvements in the ensemble performance.

We find that models pretrained on one downstream dataset often perform surprisingly well when finetuned to other downstream datasets. Notably, the downstream datasets in our study come from a wide variety of domains such as news, online forums, tweets, reviews etc (Table 1). Nevertheless, we find that pretraining on any of these downstream datasets delivers significant performance gains on most datasets (greater than half of off-the-shelf model's gains in 88% of cases) irrespective of domain. However, the best performance on a downstream dataset is usually achieved by the model pretrained on that dataset itself. Models pretrained on downstream datasets perform well on the GLUE benchmark too, despite having considerably less long-term dependencies as compared to standard upstream corpora. For example, the MNLI corpus consists of 2-sentence input texts that are concatenated in random order.

In addition to classification tasks, we also experiment with tasks such as span-based question answering, named entity recognition, and grounded commonsense inference. Self-pretraining delivers around 40-80% of the performance boost compared to models pretrained on the BookWiki corpus across ELECTRA and Roberta models. Hence, self-pretraining can perform better than fine-tuning randomly initialized models even for tasks that require prediction of more complex structured output than a single label, and for tasks whose solution relies on commonsense knowledge.

Overall, our contributions can be summarized as follows:

- Evaluation of self-pretraining across 10 downstream classification tasks and two pretraining techniques (ELECTRA and RoBERTa), with comparisons to off-the-shelf pretrained models.

- Analysis of the out-of-distribution performance of models pretrained on one downstream dataset and finetuned on other downstream datasets including the GLUE benchmark.

- Demonstration of self-pretraining's efficacy on more complex tasks than classification such as tasks requiring structured output prediction and tasks requiring commonsense reasoning.

## 2 RELATED WORK

**Self-Pretraining in Computer Vision** Most relevant to our work, two recent/concurrent works in computer vision explore self-pretraining (He et al., 2022; El-Nouby et al., 2021). In a contemporary work, He et al. (2022) showed that pretraining with a Masked AutoEncoder (MAE) objective (analogue of MLM objective for image datasets) significantly boosts the performance of ViT models on the Imagenet-1K dataset. In another related paper, El-Nouby et al. (2021) showed that pretraining solely on small-scale downstream datasets for object detection and segmentation tasks reaches the performance of Imagenet-pretrained models. Our work establishes that a similar phenomenon is observed for NLP tasks too across a wide range of datasets.

**Pretraining on Downstream Data in NLP** *Task-Adaptive PreTraining* (TAPT (Gururangan et al., 2020)) consists of taking off-the-shelf pretrained models like BERT and RoBERTa and engaging in further pretraining on the downstream datasets before finetuning them to the task at hand. TAPT has been shown to improve performance of off-the-shelf models in a variety of works (Logeswaran et al., 2019; Han & Eisenstein, 2019; Chakrabarty et al., 2019). By contrast, our work pretrains models *only* on the downstream dataset, enabling a head-to-head comparison between the performance of off-the-shelf and self-pretrained models, and (in some situations) challenging the necessity of upstream corpora altogether.

**Claims about Knowledge transfer** Many works claim that pretraining extracts generally useful *knowledge* from the upstream corpus such as linguistic patterns (Lina et al., 2019; Tenney et al., 2019; Manning et al., 2020) and facts (Petroni et al., 2019), and that this accounts for the performance gains that they enjoy on downstream tasks. Several works, e.g., in the *probing* literature (Tenney et al., 2019; Manning et al., 2020; Petroni et al., 2019), demonstrate that from the internal representations of a model, it is easy (e.g., via linear models) to predict certain linguistic features or real-world facts. However, these studies do not clarify the mechanism by which these observations relate to performance gains on downstream tasks. Tenney et al. (2019) recognizes this limitation, stating *"the observation of a (linguistic) pattern does not tell us how it is used"*. Our work suggests that to the extent that such knowledge extraction plays a role in pretraining's benefits, sufficient knowledge is often present in the downstream dataset and need not be *transferred* from huge upstream corpora.

**Challenges to the Knowledge Transfer Narrative** Multiple previous works have questioned whether knowledge transfer can fully account for the efficacy of pretraining. Improvements in performance on downstream NLP tasks have resulted from pretraining on other modalities like music and code (Papadimitriou & Jurafsky, 2020), sequences of meaningless symbols (Chiang & Lee, 2020; Krishna et al., 2021; Wu et al., 2022), and language denatured via shuffling of words (Sinha et al., 2021). On the other hand, models pretrained on language have shown improved performance on tasks dealing with other modalities such as image classification Lu et al. (2021) and reinforcement learning for games Reid et al. (2022). By contrast, we show that without surplus upstream data of any modality, self-pretraining alone can often perform comparably or even better than standard pretraining with a large upstream corpus. In a similar vein with these papers, our work suggests that a large portion of pretraining's success may come from alternative, unexplored mechanisms which have more to do with the pretraining objective than knowledge transfer from upstream corpora.

## 3 EXPERIMENTAL SETUP

Our experiments center around the ELECTRA model (Clark et al., 2019) and the RoBERTa-base model (Liu et al., 2019). On the broadest set of experiments, for which we can only afford to train one

| Dataset | Size (MB) | Classes | Domain | Task |
|---|---|---|---|---|
| AGNews (Zhang et al., 2015) | 27 | 4 | News | topic classification |
| QQP (Wang et al., 2018) | 43 | 2 | Online forum questions | paraphrase detection |
| Jigsaw Toxicity (Kaggle.com, 2018) | 59 | 6 | Wikipedia comments | toxicity detection |
| MNLI (Williams et al., 2018) | 65 | 3 | Diverse | natural language inference |
| Sentiment140 (Go et al., 2009) | 114 | 5 | Tweets | sentiment classification |
| PAWS (Zhang et al., 2019) | 139 | 2 | Wikipedia | paraphrase detection |
| DBPedia14 (Zhang et al., 2015) | 151 | 14 | Wikipedia | topic classification |
| Discovery (Sileo et al., 2019) | 293 | 174 | Web crawl | discourse marker prediction |
| Yahoo Answertopics (Zhang et al., 2015) | 461 | 10 | Online forum answers | topic classification |
| Amazon Polarity (Zhang et al., 2015) | 1427 | 2 | Product reviews | sentiment classification |

Table 1: The suite of downstream datasets used in this work along with their training set sizes

model, we employ ELECTRA because it performs better than BERT/RoBERTa given comparable compute budgets (Clark et al., 2019). In particular, we use the small variant of ELECTRA (14 million parameters), which performs similarly to BERT-base on GLUE (difference of about 2 points) while training much faster (Clark et al., 2019). However, we replicate many of these results on the larger RoBERTa-base model revealing similar results and thus establishing the generality of our findings.

During pretraining, a text sequence is fed into the model with some tokens masked out. While MLM-only models like RoBERTa only have a *generator* network that predicts the content of the masked tokens, ELECTRA has an additional discriminator module that predicts if those predictions were correct. Both the generator and the discriminator networks' parameters are updated simultaneously during pretraining. After pretraining, the generator is discarded and the discriminator is used as an encoder for finetuning on downstream tasks.

We experimented with 10 different downstream datasets (Table 1). We chose these datasets in our testbed to span different dataset sizes ranging from 27 megabytes to about 1.4 gigabytes of text in the training split. These datasets are for different tasks such as topic classification, sentiment classification, natural language inference etc., and are created using data sourced from diverse domains. Most of them are multi-class classification tasks except Jigsaw Toxicity which is a multi-label classification task, and Sentiment140 which is modeled as a regression task. For finetuning a pretrained model on any dataset, we passed the input through the model, took the vector representation of the CLS token in the final layer, and passed it through a classification head with one hidden layer to get the output.

## 4 SELF-PRETRAINING PERFORMANCE

In our first set of experiments, we compare self-pretraining's performance with other pretraining techniques. For each dataset, we pretrain an ELECTRA model on text from its training split and then finetune it on the same training data using the associated labels. To create a pretraining corpus from a downstream dataset, we concatenate the input text from each of the examples, assembling them in random order. We evaluate the performance of each finetuned model on the corresponding dataset's test split[1]. For all datasets, we evaluate performance by accuracy, except for Sentiment140 and Jigsaw Toxicity, for which we use Pearson correlation and micro-averaged AUC scores, respectively (these are not multi-class classification problems).

Notably, all self-pretrained models deliver significant performance boosts on their respective datasets (Table 2), and over half of them perform even better than the off-the-shelf model. We measured a model's *benefit* as the performance boost that it achieves over a randomly initialized model, divided by the boost achieved by the off-the-shelf ELECTRA model against the same baseline. The average benefit of self-pretraining across all datasets is 103.70%. We do not see a clear correlation between the size of the dataset and the performance of self-pretraining. For example, the highest benefit of 131.33% is achieved for the smallest dataset (AGNews), which is merely 27MB in size, while the minimum benefit is achieved on the Discovery dataset, which is the third largest dataset measuring 293MB. For each downstream dataset, we also pretrain a model on a randomly sampled subset of Wikipedia of the same size as the dataset's training corpus, and finetune it on the downstream task. This approach (called WikiSub) provides a size-adjusted comparision between using separate upstream data vs the downstream data for pretraining. We see that self-pretraining performs better

---

[1]For QQP and MNLI we just use the validation split because test set labels are private.

than WikiSub in majority of cases and when it performs worse (MNLI and Discovery datasets), the performance gap is much smaller than the gap between offshelf and self-pretrained models (Table 2).

We also evaluated the alternate pretraining technique *TAPT* as described in Gururangan et al. (2020). In this technique, we take the off-the-shelf ELECTRA model, which has already been pretrained on the upstream BookWiki corpus, and further pretrain it on the downstream dataset for 100 epochs. Self-pretraining outperforms TAPT on 6 datasets, notably including the two datasets where it outperformed the off-the-shelf models by the greatest benefit margin - *AGNews* and *Yahoo Answertopics*. Interestingly, TAPT performs worse than off-the-shelf model on the same 3 datasets where self-pretraining performs worse than off-the-shelf model (except Sentiment140). None of the three pretraining approaches seem to be uniformly better than any other.

Finally, we also evaluate the self-pretrained models on the GLUE benchmark and report results on the dev set [2]. The performance of the models on their pretraining dataset does not seem to be correlated with its GLUE score. The GLUE score also does not monotonically go up with increasing dataset size, indicating that the data domain makes some difference. For example, the Amazon Polarity corpus scores just 66.14 on GLUE despite being about 1.4 gigabytes in size, while AGNews which is 27MB in size, scores 74.30. The highest GLUE score is achieved by pretraining on Yahoo Answertopics.

| Dataset | Size(MB) | RandInit | SelfPretrain | Offshelf | Benefit% | WikiSub | TAPT | GLUE |
|---|---|---|---|---|---|---|---|---|
| AGNews | 27 | 91.75 | 94.34 | 93.72 | 131.33 | 93.51 | 94.07 | 74.30 |
| QQP | 43 | 82.93 | 90.66 | 90.34 | 104.34 | 89.16 | 90.64 | 75.43 |
| Jigsaw Toxicity | 59 | 97.83 | 98.49 | 98.53 | 94.99 | 98.35 | 98.48 | 76.65 |
| MNLI | 65 | 65.49 | 78.39 | 82.29 | 76.77 | 78.64 | 79.26 | 78.28 |
| Sentiment140 | 114 | 63.75 | 67.04 | 66.95 | 102.91 | 65.52 | 65.65 | 72.67 |
| PAWS | 139 | 50.00 | 97.53 | 97.30 | 100.49 | 97.42 | 97.85 | 74.65 |
| DBPedia14 | 151 | 98.59 | 99.22 | 99.11 | 121.17 | 99.18 | 99.23 | 70.38 |
| Discovery | 293 | 17.00 | 22.38 | 24.55 | 71.22 | 22.47 | 23.58 | 77.26 |
| Yahoo Answertopics | 461 | 61.94 | 65.26 | 64.55 | 127.31 | 64.37 | 65.05 | 79.53 |
| Amazon Polarity | 1427 | 93.86 | 96.27 | 96.13 | 106.49 | 95.82 | 96.16 | 66.14 |

Table 2: Performance of ELECTRA-small models pretrained with different techniques on various downstream datasets and on the GLUE benchmark (dev set). For reference, a randomly initialized model scores 53.20 and the off-the-shelf model scores 79.43 on GLUE.

## 5 CROSS DATASET FINETUNING

In this set of experiments, we investigated if the models pretrained on a dataset are only useful for that specific task, or are they useful across the whole spectrum of tasks that we consider. We took each model pretrained on a dataset in our testbed and finetuned and evaluated it on all other datasets in the testbed. The performance benefits provided in all cases are shown as a heatmap in Figure 2.

We found that for almost all downstream datasets, pretraining on any other dataset provides significant advantage (Figure 2). In most cases, pretraining on the downstream dataset itself performs the best. Among datasets where self-pretraining performs better than off-the-shelf model (i.e. the diagonal entry is greater than 1), pretraining on datasets of larger size does not help further. However, for the datasets where self-pretraining's benefit is much less than $100\%$ (i.e. MNLI and Discovery), pretraining on a larger dataset (e.g., Yahoo Answertopics) performs better than self-pretraining.

Among all the pretrained models, a few models perform consistently good or bad across different downstream datasets (Figure 2). For example, the model pretrained on Yahoo Answertopics gets the highest average score of 0.90 across all datasets, while the PAWS-pretrained model gives the lowest aggregate score of 0.64. Similarly, there are downstream datasets that are benefited consistently by either a large or a small margin by pretraining on different datasets. For example, performance on QQP and PAWS receives huge boosts by pretraining on most datasets. In contrast, performance on sentiment140 is low for most pretrained models, even dropping below 20% for 3 pretraining datasets.

Next, we perform an ablation to investigate that given a fixed dataset to finetune on, is it better to pretrain on the *exact* same data (i.e., using the same set of inputs), or is it better to pretrain on different data with an identical distribution. To test this hypothesis, we divided downstream datasets into two

---

[2]Following Clark et al. (2019) we exclude the WNLI task from the results.

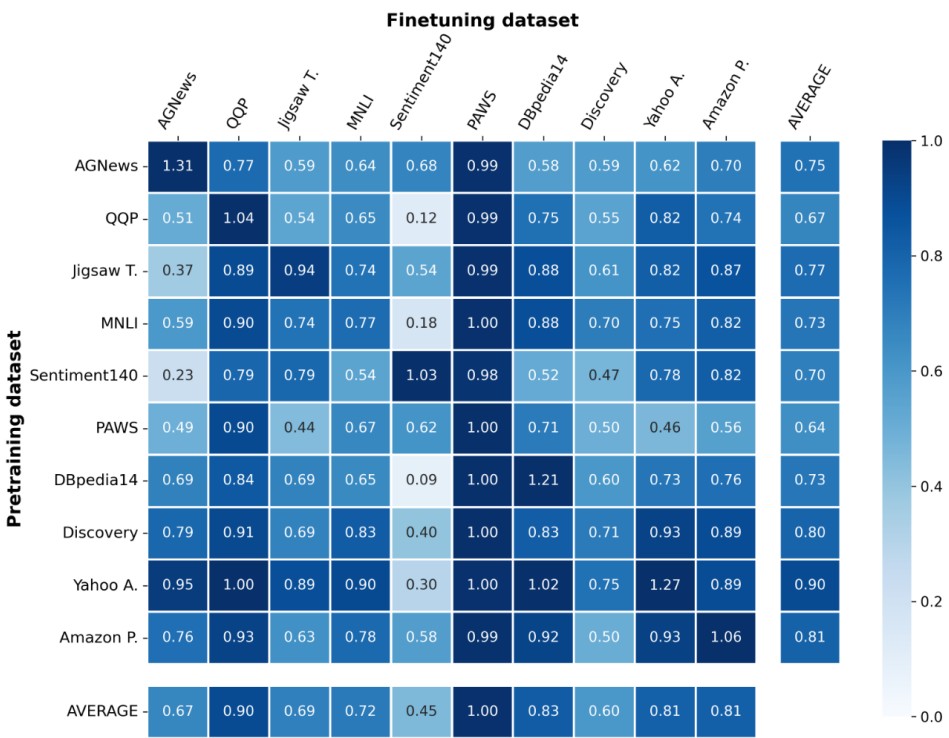

Figure 2: Performance benefits of models pretrained on each dataset, upon finetuning on each downstream dataset. Each value is the ratio of performance gains achieved by model pretrained on the row's dataset vs off-the-shelf model, relative to random initialization, upon finetuning on the column's dataset.

| | MNLI | | QQP | | Discovery | | Yahoo Answertopics | |
|---|---|---|---|---|---|---|---|---|
| | A | B | A | B | A | B | A | B |
| A | 76.00 | **76.42** | 84.28 | 84.79 | 18.78 | 18.61 | 64.18 | **64.34** |
| B | 75.93 | 75.05 | **88.73** | 88.41 | **19.99** | 19.98 | 64.09 | 64.18 |

Table 3: Performance when splitting the dataset into two equal-sized subsets A and B and then pretraining on one (row) and finetuning on another (column)

subsets (denoted as A and B) by randomly splitting the training dataset into half. We pretrained one model on each subset and then finetuned them on both subsets separately. The validation and test sets used for finetuning are the same as in the original dataset.

We do not see any consistent benefits with pretraining and finetuning on the same dataset (Table 3). Instead, we found consistent patterns where models pretrained on one split (either A or B) outperformed models pretrained on the other, irrespective of the split used for finetuning. This suggests that the pretraining data has greater influence on the final performance than the finetuning data. Additionally, we observe that finetuning the superior pretrained model, using the downstream split other than the one used for pretraining, performs the best, suggesting overall exposure to more data helps.

## 6    DIFFERENCE IN OUTPUTS OF SELF-PRETRAINED AND OFF-THE-SHELF MODELS

Since self-pretrained models and off-the-shelf models perform similarly in terms of classification accuracy, a natural question to ask is: *do these models make errors on the same set of inputs?* To answer this question, we investigate the difference in predictions made by models pretrained with different strategies across all multi-class classification tasks. In particular, given model $f_A$ and $f_B$,

| | Ensemble Accuracy | | | Error Inconsistency | | |
|---|---|---|---|---|---|---|
| Dataset | 2×SelfPretrain | 2×Offself | SelfPretrain + Offself | 2×SelfPretrain | 2×Offself | SelfPretrain + Offself |
| AGNews | 94.66 | 94.17 | 94.54 | 1.76 | 3.50 | 4.01 |
| QQP | 90.92 | 90.74 | 91.63 | 4.57 | 5.27 | 8.91 |
| MNLI | 78.51 | 82.37 | 82.31 | 6.94 | 6.42 | 14.82 |
| PAWS | 97.70 | 97.45 | 97.75 | 0.96 | 1.30 | 2.07 |
| DBPedia14 | 99.28 | 99.19 | 99.24 | 0.38 | 0.48 | 0.51 |
| Discovery | 22.98 | 25.25 | 25.02 | 7.85 | 9.18 | 12.66 |
| Yahoo | 65.32 | 64.69 | 65.64 | 5.27 | 5.49 | 9.55 |
| Amazon | 96.40 | 96.24 | 96.51 | 1.26 | 1.58 | 2.48 |

Table 4: Performance of ensemble models of self-pretrained and off-the-shelf models. For ensembling, we aggregate predictions of models after calibration with Temperature Scaling (Guo et al., 2017). We observe that in most of the datasets, SelfPretrain + Off-the-shelf ensembling does not improve over ensembles of two models with the same pre-training strategy, despite relatively higher error inconsistency of SelfPretrain + Off-the-shelf models.

we compute *error inconsistency*, defined as follows:

$$\frac{\sum_{i=1}^{n} \left( \mathbb{1}\left[ f_A(x_i) \neq y_i \wedge f_B(x_i) = y_i \right] + \mathbb{1}\left[ f_A(x_i) = y_i \wedge f_B(x_i) \neq y_i \right] \right)}{n},$$

where $\{x_i, y_i\}_{i=1}^{n}$ is the test set. Intuitively, error inconsistency captures the fraction of examples where exactly one model is correct. This definition has been commonly used to estimate diversity in model prediction (Gontijo-Lopes et al., 2022; Geirhos et al., 2020). Across all the multi-class classification tasks, in addition to computing error inconsistency between self-pretrained and off-the-shelf model, for baseline comparison, we also tabulate error inconsistency between: (i) two independently finetuned versions of a self-pretrained model; and (ii) two independently finetuned versions of the off-the-shelf model.

Compared to error inconsistency between two models with the same pretraining dataset, we observe that models trained with different pretraining datasets have high error inconsistency in predictions (Table 4). Note that for models with comparative performance, high error inconsistency highlights the high disagreement in predictions. This demonstrates that while different pretraining datasets produce similarly performing models in terms of overall accuracy, the model predictions are relatively dissimilar. Our observations here align with investigations in vision tasks, where Gontijo-Lopes et al. (2022) observed that different models trained with different pretraining datasets produced uncorrelated errors.

Since different pretraining datasets produce models with uncorrelated errors, we now seek to ensemble these models to check if uncorrelated mistakes can be converted to a correct prediction. When the models make different predictions, in particular, when one model is correct and another is incorrect, the ensemble prediction will be dominated by the model with higher confidence in their prediction. As before, we consider ensembles of (i) two independently finetuned versions of a self-pretrained model; (ii) two independently finetuned off-the-shelf models; and (iii) a finetuned version each of the self-pretrained and off-the-shelf models.

We make the following observations: First, as expected we observe that ensembling improves model performance as compared to a single model (Table 4). Second, despite having larger error inconsistency, we do not observe any significant improvements in ensembles of self-pretrained and off-the-shelf model as compared to ensembles of two models with the same pretraining strategy (Table 4). This is in contrast with findings on vision tasks where Gontijo-Lopes et al. (2022) observed that larger error inconsistency led to larger improvement in ensemble performance.

## 7    ABLATIONS WITH OTHER PRETRAINING ARCHITECTURES

We conducted our experiments so far with ELECTRA-small architecture because it is faster to pretrain than other popular models, yet delivers good downstream performance (Clark et al., 2019)

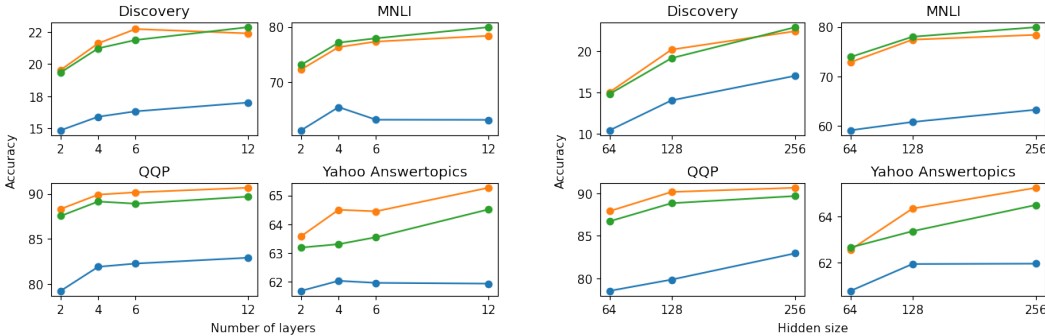

Figure 3: Variation in performance of ELECTRA models with change in number of layers and hidden size (— randomly initialized, — self-pretrained, — BookWiki-pretrained)

(e.g. comparable to BERT-base on GLUE benchmark). But we also conduct experiments with more architectures and model sizes to test the efficacy of self-pretraining more broadly.

We experiment with the RoBERTa model which uses only the masked language modeling objective, rather than ELECTRA's complex objective. We use the RoBERTa-base architecture, which has a much larger parameter count of 110 million, compared to ELECTRA-small's 14 million. Due to resource constraints, we pretrained the RoBERTa models for fewer iterations as outlined in Warstadt et al. (2020). For reference, we pretrain a RoBERTa-base model on the BookWiki corpus for the same number of iterations. Our results show again that self-pretraining performs comparably to pretraining on BookWiki corpus, delivering over 85% of pretraining benefit on 9 out of 10 datasets, and outperforming the model pretrained on BookWiki corpus (Table 5) on 5 datasets.

| Dataset | RandInit | SelfPretrain | BookWiki | Benefit % | TAPT |
|---|---|---|---|---|---|
| AGNews | 91.91 | 94.28 | 94.22 | 102.27 | 94.07 |
| QQP | 76.50 | 88.68 | 90.18 | 89.05 | 90.64 |
| Jigsaw Toxicity | 97.32 | 97.72 | 98.03 | 56.02 | 98.48 |
| MNLI | 31.82 | 75.12 | 80.90 | 88.23 | 79.26 |
| Sentiment140 | 56.68 | 68.55 | 60.19 | 338.26 | 65.65 |
| PAWS | 50.00 | 97.34 | 97.08 | 100.55 | 97.85 |
| DBPedia14 | 98.57 | 99.21 | 99.24 | 95.98 | 99.23 |
| Discovery | 17.36 | 25.85 | 26.30 | 94.91 | 23.58 |
| Yahoo Answertopics | 61.11 | 65.96 | 64.58 | 139.80 | 65.05 |
| Amazon Polarity | 89.02 | 96.68 | 96.11 | 108.13 | 96.16 |

Table 5: Performance of RoBERTa-base models pretrained with different techniques on downstream datasets.

In addition to experimenting with a *base*-sized architecture (110M parameters), we also experiment with architectures which are even smaller than ELECTRA-small. We train ELECTRA models of smaller size by either reducing the number of layers in the generator and discriminator, or reducing the hidden dimension of the discriminator[3]. As the models get smaller, self-pretraining continues to significantly outperform random initialization and often outperforms pretraining on BookWiki corpus (Figure 3). Interestingly, the relative performance of self-pretrained and BookWiki-pretrained models tends to stay the same across model size. For example, for QQP self-pretraining is always best and for MNLI BookWiki-pretraining is always best irrespective of number of layers or hidden size.

---

[3]In ELECTRA, the generator's hidden size is already much smaller than that of the discriminator by design. So we do not reduce it further, in order to have a reasonably well-performing generator.

# 8 PERFORMANCE ON STRUCTURED PREDICTION AND COMMONSENSE NLI

While the bulk of our experiments were on a variety of classification tasks, we also experiment with some tasks beyond simple classification. We experiment with three types of tasks: (i) span based question answering, (ii) named entity recognition (NER), and (iii) grounded commonsense inference. For question answering we use the SQuAD dataset (Rajpurkar et al., 2016) (v1.1) and report the F1-score. For NER, we use the CONLL-2012 NER task which uses annotations from Ontonotes v5.0 (Weischedel et al., 2013) involving 18 kinds of named entities. To measure performance, we use the overall F1 score [4]. We include SWAG (Zellers et al., 2018) and HellaSwag (Zellers et al., 2019) for multiple-choice sentence completion.

For Electra-small models, we see that for each of these datasets self-pretraining achieves more than 70% pretraining benefit, and for Roberta-base model the benefit is 40-80% (Table 6). Even for the SWAG and HellaSwag datasets, which are designed to use rely on *commonsense inference* of pretrained models, we see performance boosts by pretraining using only the task's training set.

| Datasets | Size(MB) | ELECTRA-small | | | | Roberta-base | | | |
|---|---|---|---|---|---|---|---|---|---|
| | | RI | SP | OS | Benefit% | RI | SP | BW | Benefit% |
| SQuAD | 19 | 15.82 | 63.01 | 75.96 | 78.47 | 14.93 | 67.23 | 81.89 | 78.11 |
| SWAG | 22 | 27.55 | 60.56 | 73.76 | 71.43 | 27.95 | 45.18 | 70.37 | 40.62 |
| HellaSwag | 30 | 29.27 | 39.14 | 42.91 | 72.36 | 24.53 | 31.03 | 34.28 | 66.67 |
| CONLL-2012 | 6.4 | 54.49 | 75.66 | 82.65 | 75.18 | 63.65 | 72.64 | 86.25 | 39.78 |

Table 6: Performance of ELECTRA and Roberta models pretrained with different techniques. RI: random initialization, SP: self-pretraining, OS: off-the-shelf; BW: pretrained on BookWiki by us.

# 9 CONCLUSION AND FUTURE WORK

In this work, we showed that pretraining models only on text from the downstream dataset performs comparably to pretraining on a huge upstream corpus for a wide variety of datasets. The errors made by such *self-pretrained* models on the downstream tasks are significantly different from the ones made by the *off-the-shelf* models pretrained on upstream corpora. Our results suggest that the importance of learning from surplus upstream data for improving downstream task performance may have been overestimated. Crucially, our experiments also do not show that upstream data does not help at all or knowledge transfer does not occur, but simply questions to what extent it is responsible for downstream gains. For example, the impressive zero-shot performance very large language models such as GPT-3 (Brown et al., 2020) clearly suggests knowledge transfer is involved. One direction of future work would be to investigate how the performance of self-pretraining compares of pretraining on upstream corpora as the model sizes go up by orders of magnitude.

We found that the quantity and quality of data required for pretraining to provide significant benefit (over a randomly initialized model trained only with a supervised loss) is quite low. Downstream datasets which are tiny in comparison to typical upstream corpora, still function as useful pretraining corpora for getting performance gains across a wide range of datasets. Additionally, many of these datasets (e.g. MNLI, QQP) do not contain high quality long-term dependencies and yet manage to perform well even on the GLUE benchmark for language understanding. A possible future work could be to characterize the nature of long-term dependencies in pretraining corpora and quantify their impact on downstream performance boosts.

Since self-pretraining does not involve any upstream corpus, it prevents exposure of the model to potentially undesirable contents in the large upstream corpus, while still delivering large performance benefits. Research has demonstrated the negative influence of web-sourced pretraining corpora on models, such as generating toxic language (Gehman et al., 2020) or reflecting racial biases in predictions (Ahn & Oh, 2021). For use cases that require avoding such issues, self-pretraning can provide a viable alternative to standard pretraining. In future work, we hope to compare how self-pretrained models and off-the-shelf models perform on these negative measures such as toxicity and social biases.

---

[4]We use the seqeval library for evaluation (https://github.com/chakki-works/seqeval)

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

# A APPENDIX

## A.1 THE ROLE OF SENTENCE ORDER IN PRETRAINING CORPORA

For virtually all pretrained models like BERT, ELECTRA, XLNet, the sentences in the pretraining corpora are ordered as they naturally occur in some document such as Wikipedia article. Devlin et al. (2019) mention in their work : *"It is critical to use a document-level corpus rather than a shuffled sentence-level corpus (...) in order to extract long contiguous sequences."* However, for many of our pretraining corpora made from downstream datasets, the sentence taken in order do not form a coherent document or narrative text. For example, in the MNLI or QQP corpora, neighboring sentences will simply be premise-hypothesis pairs or potential paraphrase candidates.

Despite the sentence order not forming a coherent document, many pretraining corpora achieve high performance boosts on the GLUE language understanding benchmark (Table 7). For example, MNLI achieves around 96% of the performance boost of the off-the-shelf model (Table 7). Interestingly, shuffling the sentences in these corpora leads to a large drop in performance (Table 7). This suggests that there is some value to keeping the sentence order in a way that puts sentences from the same example in datasets like MNLI and QQP next to each other. A likely explanation of this is in Levine et al. (2021) where authors showed that including similar sentences in the same input sequence when pretraining should lead to improved performance via theoretical analysis and empirical experiments.

We test if GLUE performance can be improved by artificially re-ordering a set of sentences to promote the occurrence of similar sentences together. We rearrange the sentences in the sentence-shuffled versions of pretraining corpora to encourage content overlap among neighboring sentences, and see if this can recover some of the drops in performance that occurred due to shuffling. Our algorithm creates the corpus by iteratively appending sentences to it, such that at each step the new sentence is the one with maximum TF-IDF similarity with the previous sentence. Such a way of constructing a corpus by similarity based retrieval has been used in past works (Levine et al., 2021; Yao et al., 2022), with the main difference that they retrieved sentences from external corpora similar to the ones present in the downstream dataset, whereas we simply use it to reorder sentences already present in the downstream dataset for pretraining We also make sure that the algorithm does not accidentally recover the original order of sentences (e.g. by matching the premise-hypothesis pairs originally in the MNLI dataset).

We experiment with 5 different datasets and find that the sentence-reordering scheme improves performance compared to random sentence order for all of them except QQP. For Discovery and DBPedia14 datasets, it scores even higher than our *standard* sentence ordering scheme which preserves the adjacency and order of sentences within each datapoint. This shows that re-ordering sentences to promote content similarity between neighboring sentences, can potentially improve GLUE score, without introducing any new information or narrative structure.

## A.2 IMPLEMENTATION DETAILS FOR PRETRAINING AND FINETUNING

**Hyperparameters for pretraining**  For pretraining ELECTRA-small models, we use the standard hyperparameters (Table 8) as described in Clark et al. (2019). For the Roberta-base models, training

| Pretraining Dataset | Random | Standard | TF-IDF(Ours) |
|---|---|---|---|
| None (RandomInit) | - | 53.20 | - |
| Sentiment140 | - | 72.67 | 75.29 |
| DBpedia14 | 72.82 | 70.38 | 75.44 |
| Discovery | 71.79 | 77.26 | 78.94 |
| MNLI | 62.80 | 78.28 | 76.33 |
| QQP | 71.09 | 75.43 | 69.57 |
| BookWiki (Off-the-shelf) | - | 79.43 | - |

Table 7: GLUE scores achieved by different strategies for ordering sentences from the downstream dataset used for pretraining. Random: randomly ordered sentences; Standard: sentences within a datapoint occur contiguously in original order; TF-IDF: sentences reordered using content similarity.

with the standard hyperparameters with our computing resources would be prohibitively slow, and so we used hyperparameters from Warstadt et al. (2020) which require lesser time to train (Table 8). For task-adaptive pretraining(TAPT), we follow Gururangan et al. (2020) and further pretrain off-the-shelf models for 100 epochs on the downstream task's training set, with the first 6% of the resulting total updates used for learning rate warmup.

**Hyperparameters for finetuning**  For finetuning the models on the 10 downstream datasets, we use hyperparameters as shown in Table 9. We use the AdamW optimizer (Loshchilov & Hutter, 2018) for finetuning. We use early stopping based on validation set performance. The validation metric used is mean squared error for the sentiment140 dataset (regression), average binary crossentropy for the jigsaw dataset (multi-label classification), and accuracy for all other datasets (multi-class classification). The patience parameter for early stopping is set to 3 epochs. For finetuning ELECTRA-small models on the GLUE datasets, we use the standard learning rate of 1e-4 following Clark et al. (2019).

**Details about use of downstream datasets**  All downstream datasets used in this paper were sourced from the Huggingface library[5]. For the Yahoo Answertopics dataset, we use only the text from the answer (not the question) as input to the models (both for pretraining and finetuning). For the PAWS dataset, we use the version called "Unlabeled PAWS$_{wiki}$" in Zhang et al. (2019), which is actually *not* unlabeled but has silver labels. We preferred that version over others because of its larger size. For datasets which had a train and test split but no validation split (e.g. Yahoo Answertopics), we extracted 5000 random datapoints from the the train split to make the validation split. If a dataset had a train and validation split but no test split (e.g. Unlabeled PAWS$_{wiki}$), we designated the validation split to be the test split, and created a new validation set by extracting 5000 random datapoints from the train set.

| Hyperparameter | ELECTRA | Roberta |
|---|---|---|
| Size (Parameter count) | Small (14M) | Base (110M) |
| Training steps | 1M | 100K |
| Warmup steps | 10K | 6K |
| Batch size | 128 | 512 |
| Peak learning rate | 5e-4 | 5e-4 |
| Sequence length | 128 | 512 |

Table 8: Hyperparameters used for pretraining models

| Hyperparameter | ELECTRA | Roberta |
|---|---|---|
| Training epochs | 20 | 20 |
| Batch size | 32 | 32 |
| Learning rate | {1e-4,1e-5} | 2e-5 |
| Max sequence length | 512 | 512 |

Table 9: Hyperparameters used for finetuning models on 10 downstream tasks

## A.3 SOFTWARE PACKAGES AND HARDWARE USED

For pretraining ELECTRA models, we used Nvidia's implementation of the ELECTRA codebase [6], run using Nvidia's Tensorflow cotainer image 21.07 [7]. For pretraining Roberta models, we used the official implementation in the Fairseq library[8]. For finetuning experiments, we used the AllenNLP library for training and evaluation routines, coupled with the Huggingface library for the model architectures.

---

[5]https://huggingface.co/docs/datasets/index
[6]https://github.com/NVIDIA/DeepLearningExamples/tree/master/TensorFlow2/LanguageModeling/ELECTRA
[7]https://docs.nvidia.com/deeplearning/frameworks/tensorflow-release-notes/rel_21-07.html
[8]https://github.com/facebookresearch/fairseq

We used a collection of Nvidia V100 (32GB) and A6000(48GB) GPUs for our experiments. Pretraining an ELECTRA-small model takes around 1.5 days on 2 GPUs while pretraining a Roberta-base model takes around 1.5 days on 4 GPUs.

