# OpenReview forum: "Downstream Datasets Make Surprisingly Good Pretraining Corpora"
_ICLR.cc/2023/Conference — Submitted to ICLR 2023_

### Official Review · Reviewer_fKqx · 2022-10-17

**Confidence:** 4
**Correctness:** 2
**Technical Novelty And Significance:** 3
**Empirical Novelty And Significance:** 3
**Recommendation:** 5

**Clarity, Quality, Novelty And Reproducibility:**

The paper is overall very clear. The results are novel to the best of my knowledge and seem quite reproducible.


**Strength And Weaknesses:**


Strengths:
- The presented findings are very interesting, and are important for questioning some of the basic assumption of the field of NLP in recent years: the need of huge, typically web-crawled text corpora for training MLMs. These results can help in mitigating some of the biggest problems of the field nowadays: training on huge corpora which potentially contains biases and toxic language, and in general makes training extremely expensive in terms of energy, time and money.

- The authors present an extensive set of experiments, showing that the their findings translate to different tasks, to transfer learning, and to different architectures.

Weaknesses:
- Despite the very impressive results, I think the elephant in the room, which I was surprised not to see any mention of in the paper, is how much do the results have to to do with the corpus being one of the downstream task corpus, or simply a smaller corpus. That is, a baseline of pretraining on a general pretrain corpus of similar size to the downstream corpora is missing and is very important to shed light on the results here. Is the conclusion from these results basically that you need far less pre-training in order to get good finetuning performance? I am aware that size doesn't explain it all, but I am also far from convinced that the corpora being downstream corpora makes an important difference here.

- The error consistency experiments are interesting, but raise a few potential concerns. First, computing the prediction difference should be done on the validation set, not the test set (especially given the attempts to use this information to do smart ensembling, which could lead to overfitting the test set). Second, there is no evidence presented in this paper that using the model confidence to select the correct model should lead to good ensemble performance. Are models more confident in their correct predictions compared to their wrong predictions? This is a general desired property of models, but not one that is guaranteed and in particular not one that is explored in this paper (at least not reported), which makes it it not very surprising that these experiments failed. Finally, I am not sure I agree with the conclusion (stated from the conclusion section): "The errors made by such self-pretrained models on the downstream tasks are *significantly different* from the ones made by the off-the-shelf models pretrained on upstream corpora." The error inconsistency numbers are indeed higher among the two approaches, but still overall quite low in 4/8 of the datasets, and even the higher ones are only around 10%.

- The corpus reordering experiment (Sec. 7) is interesting, and points to the importance of batching similar documents during pretraining. Nonetheless, it does not seem related to long-term dependencies (as suggested by the title and the intro of that section), as even the authors do not argue that such dependencies exist in the reordered corpus.

- Finally, more a suggestion than a weakness: what about zero/few-shot? The assumption this paper is trying to break, as far as I understand it, is that pretraining does not lead to knowledge transfer. This assumption stands in the heart the strong few-shot/zero-shot results observed in recent years, even with reasonably sized models (e.g., [1,2]). Did the authors consider running such experiments with their minimally pre-trained models? I agree that this might be out of scope for this paper, but something the authors could consider.

[1] https://arxiv.org/abs/2001.07676
[2] https://arxiv.org/abs/2104.05240


**Summary Of The Paper:**

This paper follows a line of papers who question the importance of massive corpora for pre-training MLMs. The authors show that by pre-training MLMs on downstream corpora (which are far smaller than standard pre-training corpora), they reach similar performance on the corresponding downstream task to that of off-the-shelf MLMs. Surprisingly, the authors also show that these models also transfer quite well to other tasks. They then show that model prediction of models trained on these downstream corpora is typically quite different compared to standard models (though this does not translate to strong ensemble results).


**Summary Of The Review:**

The findings in this paper are quite interesting and could be of value to the community. However, many of the central claims in this paper are not validated, e.g., the importance of the downstream corpora, the _significant difference_ in the error patterns between the different approaches, and the argument regarding long-term dependencies.

---

> ### Author Response · Authors · 2022-11-16
> **Response to Reviewer fKqx**
>
> We thank you for your detailed and thoughtful comments.
>
> > **... how much do the results have to do with the corpus being one of the downstream task corpus… That is, a baseline of pretraining on a general pretrain corpus of similar size to the downstream corpora…. Is the conclusion from these results basically that you need far less pre-training in order to get good finetuning performance?.**
>
> Following your suggestion, for each downstream dataset, we pretrained an ELECTRA model using a random subset of Wikipedia equal in size to the downstream training set. We then fine-tuned that pretrained model on the downstream dataset. We have added the resulting performance  in a new column in Table 2. In 8 cases, self-pretraining outperformed pretraining on a size-matched upstream corpus and in 2 cases, it performed worse.
> Hence, pretraining on exactly the downstream dataset on which we have to finetune can often perform better than using upstream data of comparable size from a large general corpus such as Wikipedia.
>
> We would like to add that our primary argument is not about the size of pretraining corpus but about its source. Concretely, our most important contribution is to show that pretraining can be surprisingly effective, even when no additional data, beyond that already available for finetuning the downstream predictive model is incorporated.
> .
>
> > **The error consistency experiments are interesting, but raise a few potential concerns. First, computing the prediction difference should be done on the validation set, not the test set (especially given the attempts to use this information to do smart ensembling, which could lead to overfitting the test set).**
>
> We apologize for confusion here. For ensembling, we simply take the average of model prediction probability after calibration on validation. Since, we do not make any choice based on peeking at test performance, we believe that we avoid overfitting to the test set.
>
> > **Second, there is no evidence presented in this paper that using the model confidence to select the correct model should lead to good ensemble performance... Are models more confident in their correct predictions compared to their wrong predictions? This is a general desired property of models, but not one that is guaranteed … which makes it not very surprising that these experiments failed.**
>
> We agree that in general it is not guaranteed that models will be more confident in their correct predictions compared to their wrong predictions. Our aim with experiments in Section 6 was to understand differences in model behavior with different training strategies. In particular, our ensembling experiments were motivated by recent findings in Gontijo-Lopes et al. (2022) where authors observed that that larger error inconsistency in vision models (due to different training strategies) led to larger improvement in ensemble performance. This was in contrast to our findings where despite having larger error inconsistency, we do not observe any significant improvements in ensembles of self-pretrained and off-the-shelf model as compared to ensembles of two models with the same pretraining strategy.
>
> Gontijo-Lopes et al. “No One Representation to Rule Them All: Overlapping Features of Training Methods” ICLR 2022
>
> > **Finally, I am not sure I agree with the conclusion "The errors made by such self-pretrained models on the downstream tasks are significantly different from the ones made by the off-the-shelf models pretrained on upstream corpora." The error inconsistency numbers are indeed higher … but still overall quite low in 4/8 of the datasets, and even the higher ones are only around 10%.**
>
> Note that here error inconsistency is being compared between two high accuracy models. Specifically, error inconsistency number should be compared on the scale of 0 to 2*E where E is approximately the error of the individual model. This is because 2*E is approximately the upper limit of error inconsistency. Hence 10% error inconsistency numbers can be significant when individual error of the models is around 20%.
> For datasets where the error inconsistency numbers are relatively smaller, the error of the individual model is also smaller by a similar order of magnitude.

---

> > ### Author Response · Authors · 2022-11-16
> > **Response to Reviewer fKqx (continued)**
> >
> > > **Finally, more a suggestion than a weakness: what about zero/few-shot? The assumption this paper is trying to break, as far as I understand it, is that pretraining does not lead to knowledge transfer. This assumption stands in the heart the strong few-shot/zero-shot results observed in recent years, even with reasonably sized models (e.g., [1,2]). Did the authors consider running such experiments with their minimally pre-trained models? I agree that this might be out of scope for this paper, but something the authors could consider.**
> >
> > We want to be very clear at the outset that we are not claiming that genuine “transfer” has no role to play in pretraining’s success. Rather, our exciting finding is that a significant portion (but not all) of pretraining’s benefits can be realized even in the absence of additional data.
> >  To stress the point further, transfer clearly plays a role in many of the exciting new developments in NLP. For instance, the zero-shot model performance of large language models such as GPT-3 can only be due to transfer. Our more subtle claim is that pretraining has significant benefits that cannot be explained by cross-dataset transfer. Even when we pretrain using only the exact same data on which finetuning will be performed, we realize (sometimes transformative) benefits in  downstream performance compared to a non-pretrained model. Moreover, for several classification tasks, such self-pretrained models perform even better than off-the-shelf models trained on a huge amount of external data (Bookwiki corpus).
> >
> >
> > > **The corpus reordering experiment (Sec. 7) is interesting, and points to the importance of batching similar documents during pretraining. Nonetheless, it does not seem related to long-term dependencies (as suggested by the title and the intro of that section), as even the authors do not argue that such dependencies exist in the reordered corpus.**
> >
> > Thanks for your comment. We have changed the title and content of that section (now Section A.1 in the Appendix) to clarify that the results reflect the impact of reordering sentences to keep similar sentences together in the pretraining corpora, rather than long-term dependencies. We have also added a suggested reference (Levine et al. 2022) which provides theoretical justification for the improved performance by keeping similar documents together while pretraining.
> >
> > Levine et al. “The inductive bias of in-context learning: Rethinking pretraining example design.” ICLR 2021

---

> > > ### Comment · Reviewer_fKqx · 2022-11-22
> > > **Thank you for your efforts and your detailed response**
> > >
> > > Table 2 sheds some light about my concern. I am afraid it further amplifies it. My interpretation of it is that pretraining on a much smaller corpus closes almost all the gap between random init and full pretraining. The rare cases where it doesn't (e.g., MNLI), neither does the proposed approach. In most cases the gaps between WikiSub and SelfPreTrain are wihtin 0.5%, and in all cases within 1.5%. This suggests that the gist of the findings is mostly around the redundant size of the pretrain corpus, and much less about the data being the downstream data.
> > >
> > > To conclude, I think this paper is interesting and has great potential, but while I realize that the primary argument the authors are trying to make "is not about the size of pretraining corpus but about its source", I argue that their results actually relate more to the size than to the source. This is not to say that these results are not interesting, but they requite a somewhat different framing and comparison with previous work.
> > >
> > > I am therefore keeping my score (5).

---

> > > > ### Author Response · Authors · 2022-12-08
> > > > **Thank you for your reply**
> > > >
> > > > Thanks for your reply. We want to make just one clarification. While we agree that the accuracy difference between self-pretraining and WikiSub is small (0.5 to 1.5%), the relative accuracy difference over randomly initialized models is significant in the percentage points. We have included WikiSub in the heatmap here (https://ibb.co/CBcgT8P) in the cross fine-tuning table. Additionally, self-pretraining is consistently better than WikiSub pretraining on 8 out of 10 datasets. This also highlights that often performance gains attributable to pretraining are driven primarily by the pretraining objective itself and are not always attributable to the use of external pretraining data.

---

### Official Review · Reviewer_XBVs · 2022-10-21

**Confidence:** 4
**Correctness:** 3
**Technical Novelty And Significance:** 2
**Empirical Novelty And Significance:** 3
**Recommendation:** 6

**Clarity, Quality, Novelty And Reproducibility:**

- I found this paper to be quite clear and readable.
- I do think the results are somewhat surprising and novel, although there's a bit of overclaiming (since I suspect that these results are largely due to the similarity of the downstream datasets).
- The work seems reproducible, with details in appendix A about the pre-training and fine-tuning process.

**Strength And Weaknesses:**

Strengths:
-  Interesting research question
- Thorough experimental setup, a variety of text classification datasets examined. Compared against reasonable baselines (e.g., random initialization).

Weaknesses:
(1) It's not clear that the results answer the question of "Our results suggest that in many scenarios, performance gains attributable to pretraining are driven primarily by the pretraining objective itself and are not always attributable to the incorporation of massive datasets"
  - What happens if you sample upstream corpora to match the size of the downstream dataset, and then compare pre-training on downstream vs upstream with similar data sizes?

(2) Related to (1), these datasets are all text classification datasets that are pretty similar, so it's somewhat unsurprising to me that pre-training on downstream corpora works well when fine-tuning on other datasets. Furthermore, for many of the benchmarks, the performance of RandomInit is already quite high and the gains from pre-training are pretty small already, so maybe it's not so surprising that pre-training on _anything_ helps.
  - I think the analysis of model size is quite helpful, since a natural question is whether or not these results are due to the ELECTRA small model being too small to benefit from pre-training on massive upstream datasets. But another concern in this vein that these downstream datasets are too simple to really see many gains from pre-training upstream datasets.
  - It'd be nice to see performance on other types of downstream datasets, like QA. For instance, even looking like something like MNLi performance, it seems like performance is comparable to upstream datasets when pre-training on similar sentence-pair datasets (e.g., QQP or PAWS), but is far lower when pre-training on other downstream datasets that aren't as similar.

Questions:
- Why is the performance of PAWS on RandomInit 50%? Is this a coincidence? I'm a bit surprised that it's so low.

**Summary Of The Paper:**

This paper conduct experiments on pre-training on downstream datasets, rather than the traditionally-used large upstream datasets. The authors explore a variety of text classification settings, showing that pre-training on the downstream text classification dataset yields results on the downstream text classification dataset that are competitive with the traditional upstream pre-training corpora.

This paper then evaluates whether these datasets trained on downstream datasets end up being overly downstream-specific, or if they can still provide strong performance when fine-tuned on other downstream corpora. Surprisingly, the authors find that pre-training on one downstream dataset can still provide benefits on other downstream datasets over random initialization.

Furthermore, the work evaluates whether these models trained on upstream vs downstream datasets make similar errors. Although their errors are not necessarily correlated, ensembling the models does not lead to substantial gains.

Lastly, the authors explore a variety of ways of trying to better leverage downstream datasets for pre-training, including some heuristics for increasing GLUE score by increasing the length of dependencies.

**Summary Of The Review:**

I think this paper conducts an interesting analysis and points out a conclusion that makes sense in hindsight, but I don't think many would have realized a-priori. However, I do think that this conclusion could be better-qualified (e.g., this is not a general conclusion, but one about text classification datasets). Furthermore, it's not clear to me that this paper directly answers the question of whether pre-training data scale or pre-training objective is the key to better transfer performance. Despite this, I think the work is thorough and interesting and would be of interest to the ICLR community.

---

> ### Author Response · Authors · 2022-11-16
> **Response to Reviewer XBVs**
>
> Thank you for your review of our work. We are pleased to see that you found our analysis interesting and thorough. We have incorporated many of your suggestions in our revised manuscript which we believe has made our draft better.
>
> > **“It'd be nice to see performance on other types of downstream datasets, like QA. ”**
>
> While our previous manuscript had experiments with only classification tasks, in the new manuscript we have added experiments with 4 new datasets dealing with structured prediction and commonsense inference tasks. These datasets are:
>
> 1. SQUAD (Question answering)
>
> 2. CONLL-2012 (Named entity recognition)
>
> 3. SWAG (grounded commonsense inference)
>
> 4. HellaSwag (grounded commonsense inference)
>
> We have added the results for Electra-small and Roberta-base models in Table 6. We find that for Electra-small, self-pretraining achieves greater than 70% of the (pretraining) benefit achieved by an off-the-shelf model on all datasets. For Roberta-base, self-pretraining achieves 40-80% of the benefit of a model trained on large-scale upstream corpus, depending on the dataset. These results show that a significant portion of pretraining’s benefits can be realized in the absence of transfer (from an upstream dataset). Notably, the gap between self-pretraining and off-the-shelf performance also leaves a significant role for transfer to play.
>
> > **”It's not clear that the results answer the question of "Our results suggest that in many scenarios, performance gains attributable to pretraining are driven primarily by the pretraining objective itself and are not always attributable to the incorporation of massive datasets"”**
>
> We would like to clarify what we mean when claiming that “in many scenarios, performance gains … are not always attributable to the incorporation of massive datasets", we meant massive *external* datasets (upstream). Our primary finding is about the source of the pretraining data rather than its size. We found that it is not essential to use text from external sources (e.g., Wikipedia) for pretraining, and even pretraining using the text from downstream training set only (self-pretraining) can provide significant benefit in many cases. Self-pretraining can also sometimes perform even better than pretraining on a massive external corpus like BookWiki. We have edited the abstract to clarify the phrasing of our claim.
>
> > **”What happens if you sample upstream corpora to match the size of the downstream dataset, and then compare pre-training on downstream vs upstream with similar data sizes?”**
>
>  We have updated our manuscript to include the following additional experiments:  for each downstream dataset, we pretrained an ELECTRA model using a random subset of Wikipedia equal in size to the downstream training set. We then fine-tuned that pretrained model on the downstream dataset. We have added the resulting performance numbers as a column in Table 2. In 8 cases, self-pretraining outperformed pretraining on a size-matched upstream corpus and in 2 cases, it performed worse.
>
> > **”these downstream datasets are too simple to really see many gains from pre-training upstream datasets”**
>
> We collected the set of downstream datasets to cover a vast span of dataset sizes (from 27MB to 1.4GB), and it happened to be a mixture of tasks of varying difficulty. However, for more than half of the datasets, the performance of randomly initialized models is less than 90% for both Electra-small and the larger Roberta-base model, signifying that they are not that easy to solve even without any pretraining. For every single dataset and for both pretraining architectures, we see that self-pretraining consistently outperforms random initialization, and for 10 out of 20 cases (Roberta and Electra models applied to each of the 10 datasets), we see an increase of more than 5 points in performance (Table 2 and 5). This demonstrates the magnitude of benefit that self-pretraining can provide on these tasks.
>
> In the revised manuscript, we have also added new experiments with structured prediction and commonsense tasks (Table 6) where self-pretraining provides even higher gains (e.g., an increase of over 50 points in F1-score for question answering on SQuAD).
>
> > **”whether or not these results are due to the ELECTRA small model being too small to benefit from pre-training on massive upstream datasets”**
>
> In addition to the small Electra model, we also used the larger Roberta-Base model which gives increases performance by more than 5 points compared to random initialization in 6 out of 10 tasks, and provides >90% of the pretraining benefit on 7 tasks (Table 5), suggesting that even for moderately-sized models self-pretraining can help significantly.

---

> > ### Comment · Reviewer_XBVs · 2022-11-28
> > **response to authors**
> >
> > Thanks for the updated results, I really appreciate it. I think this does help me better understand where these results might be coming from, but it's a bit hard for me to understand how general this conclusion really is / what the ramifications are (e.g., in light of the scale of the models examined). I think this work is reasonably well-executed and could be of interest, but I'm not sure the conclusions are clear enough to warrant an increase in score to 7.

---

### Official Review · Reviewer_i9wP · 2022-10-24

**Confidence:** 4
**Correctness:** 3
**Technical Novelty And Significance:** 2
**Empirical Novelty And Significance:** 2
**Recommendation:** 5

**Clarity, Quality, Novelty And Reproducibility:**

The paper is well written and presented, but limited in terms of novelty and originality.

**Strength And Weaknesses:**

Strengths:
* The paper is well written and easy to follow.
* The authors conduct a thorough empirical study of pre-training on 8 downstream datasets. During fine-tuning, they compare their models to random initialization, off-the-shelf pre-trained models and task-adaptive pre-training. The authors find that in most cases the downstream datasets can also serve as a good task-specific pre-training datasets.
* While most of the paper focuses on a small ELECTRA model, the authors consider how the results transfer to a larger RoBERTa-bert model, and show that similar observations can be made.

Weaknesses:
* In the introduction the authors make the strong claim that “[...] the benefits from pretraining in this case cannot be attributed to knowledge transfer”, while in Section 2 the authors are more nuanced and allow for the possibility that “sufficient knowledge is [...] present in the downstream dataset”. It seems possible that the models can acquire linguistic and factual knowledge from these datasets during pre-training, which cannot be learnt efficiently from only predicting the downstream task labels. Since “knowledge transfer” arguments are concerned primarily with the type of knowledge transferred, rather than the source of knowledge, the authors would have to show that certain probing results cannot be replicated in their pre-trained models to support the claim from the introduction.
* The observation that small task-specific pre-training corpora can bring strong benefits for a particular fine-tuning task was already made by Yao et al., 2022, which is not referenced in the paper. It is also relevant to Section 7, as Yao et al. construct examples by concatenating top-k BM25 retrieved documents, which is similar to using TF-IDF. Another relevant reference in Section 7 is Levine et al., 2022, who propose a similar retrieval heuristic for constructing better pre-training examples.
* The name “self-pretraining” seems unfortunate since (a) it evokes parallels to self-training, where a model is used to generate pseudo-labels (see Amini et al,. 2022), and (b) confounds general, task-agnostic pre-training with task-specific pre-training.
* Regardless of the RoBERTa-base experiments, it remains unclear how the results transfer to auto-regressive models, and how relevant they are to the web-scale pre-training of very large language models. While experiments would be unrealistic here, I would appreciate a fairer and more in-depth discussion in the paper.


Yao et al., NLP From Scratch Without Large-Scale Pretraining: A Simple and Efficient Framework, ICML 2022

Levine et al., The Inductive Bias of In-Context Learning: Rethinking Pretraining Example Design, ICLR 2022

Amini et al., Self-Training: A Survey, 2022


**Summary Of The Paper:**

The paper investigates the use of downstream datasets as pre-training corpora and finds that for small ELECTRA models, as well as RoBERTa-base, downstream datasets are competitive to large-scale general-purpose pre-training corpora. Furthermore, some downstream datasets (esp. Yahoo Answers) are good pre-training corpora for most considered downstream tasks. Experiments show that their pre-trained models tend to make different errors than off-the-shelf models after fine-tuning. Furthermore, the authors find that sentence contiguity is important when pre-training on sentence-pair datasets, and experiment with using TF-IDF to construct better pre-training examples.

**Summary Of The Review:**

While this paper features a series of careful experiments, I do not think that the findings are particularly surprising or actionable, and I have some concerns about its claims.

---

> ### Author Response · Authors · 2022-11-16
> **Response to Reviewer i9wP**
>
> Thank you for the constructive feedback and suggestions for our work. We were happy to see that you found our empirical experiments thorough. We have incorporated many of your proposed suggestions and believe that they have significantly improved the quality of the draft.
>
> > **”authors make the strong claim that “[...] the benefits from pretraining in this case cannot be attributed to knowledge transfer” …  It seems possible that the models can acquire linguistic and factual knowledge from these datasets during pre-training”**
>
> We would like to clarify what we mean when claiming that the benefits of pretraining, when done on the downstream dataset itself, cannot be attributed to “knowledge transfer”.
> Our statement is about the “transfer” part of the phrase, which we take to specifically indicate a “transfer” (of some learned pattern or other object) across datasets. While pretraining can help to extract information (that might reasonably be called “knowledge”) from a dataset, it is not “transfer” in the precise sense indicated above. If we were to begin to label techniques that make sense only of the task-specific downstream data as “transfer learning” then it would likely become difficult to draw the line for what precisely constitutes transfer learning. In short, we do not take on the question of what precisely constitutes “knowledge” or what types of knowledge (or more generally, patterns) are learned/extracted during pretraining but address only the extent to which “transfer” is required to realize the familiar benefits.
>
> We have edited the manuscript in an attempt to make this point clearer and hope that this explanation and the associated edits address your concerns.
>
>
> In this work, we do not attempt to characterize the features learnt by the models during pretraining ie we do not check whether they are learning linguistic or other kinds of knowledge. How one might characterize the linguistic knowledge present in pretrained models (e.g., via probing) is lively and hotly debated research topic (Maudslay et al 2021, Niu et al 2022) but lies  outside the scope of our paper.
>
> Maudslay et al. “Do Syntactic Probes Probe Syntax? Experiments with Jabberwocky Probing”, NAACL21
>
> Niu et al. “Does BERT Rediscover a Classical NLP Pipeline?”, COLING22
>
> > **”The observation that small task-specific pre-training corpora can bring strong benefits for a particular fine-tuning task was already made by Yao et al., 2022”**
>
> Thank you for bringing this work to our attention! In the updated manuscript, we mention and cite this work in the Introduction section.  We note that while Yao et al does pretrain with a small amount of data, they still get text for pretraining from external sources other than the downstream train set. In contrast, our primary contribution is showing that we can get large benefits *without* using any external data apart from the data already available in the downstream training set. Additionally, there is also some methodological difference in pretraining with Yao et al who do use the MLM pretraining objective and the downstream fine-tuning objective together in a multitasking setup.
>
> > **”Another relevant reference in Section 7 is Levine et al., 2022, who propose a similar retrieval heuristic for constructing better pre-training examples.”**
>
> Thank you very much for sharing this reference which is quite relevant to our results on ordering sentences from downstream corpora. We have added a reference to this paper in the section  about reordering corpus sentences. Levine et al provides a theoretical explanation of our results by showing why keeping similar sentences together in a pretraining example can improve performance. We also note that while Levine et al uses retrieval heuristic to fetch additional data from an external corpus for pretraining, we simply use it to reorder the sentences already present in the downstream dataset before pretraining, without involving any external data source.

---

> > ### Author Response · Authors · 2022-11-16
> > **Response to Reviewer i9wP (continued)**
> >
> > > **”it remains unclear how the results transfer to auto-regressive models, and how relevant they are to the web-scale pre-training of very large language models”**
> >
> > We agree that in the case of very large scale autoregressive models such as GPT-3 trained on much larger corpora than BookWiki, some kind of knowledge transfer certainly plays a major role, especially given their zero-shot performance. We have added a discussion on this in the Conclusion section of the paper. While we lack the compute resources to experiment with self-pretraining at larger model sizes, even the results we have at the current scale hold both conceptual relevance for understanding the mechanism by which pretraining works, and practical relevance in domains where using enormous models is not feasible due to compute/energy restrictions like usage on mobile devices.
> > Self-pretraining can also be useful for users who do not want to use models pretrained on web-scale data because its predictions can be influenced by unknown and potentially harmful aspects of external data such as racial or gender biases. While the default choice in such cases would be to use a randomly initialized model, we show that one can get much better performance by simply pretraining on the finetuning data itself.

---

> > > ### Comment · Reviewer_i9wP · 2022-11-18
> > > **Response to Authors**
> > >
> > > I am glad to see the revisions. I appreciate the added citations and the more nuanced claims about knowledge transfer in the introduction. They address most of my concerns that readers might take away a misleading answer to the complex question of the role of linguistic/factual knowledge in pre-trained models.
> > >
> > > I have raised my score to 5 to reflect the updated manuscript, but I'm still leaning negative due to limited technical novelty and the central claim of "surprising-ness" being anticipated by Yao et al., despite minor methodological differences.

---

### Official Review · Reviewer_8VQm · 2022-10-25

**Confidence:** 4
**Correctness:** 4
**Technical Novelty And Significance:** 3
**Empirical Novelty And Significance:** 4
**Recommendation:** 8

**Clarity, Quality, Novelty And Reproducibility:**

The paper is easy to follow and the writing is clear. A comprehensive investigation of self-pretraining is (to my knowledge) novel in the NLP literature, and the related work is addressed in the paper.

**Strength And Weaknesses:**

Strengths:
- Experiments include a variety of text domains and dataset sizes, and the core findings of the paper hold across all data conditions
- Both off-the-shelf and randomly-initialized models are included as points of comparison in the experiments, providing appropriate context for interpreting the effectiveness of self-pretraining
- Understanding the reasons underlying the effectiveness of pre-training can have far-reaching implications given the current state of the field. Providing concrete evidence that casts doubt on the popular claims of "knowledge transfer" contributes to a experimentally-grounded understanding of the topic.

Weaknesses:
- The ensembling results appear to have ensembling take place over probabilities, rather than discrete predictions -- at least that appears to be the case given that only two models are being ensembled. Is there any way to apply majority-vote style ensembling instead, to get around the issues with uncalibrated model confidence levels?
- The discussion on how to construct artificial long-term dependencies is very interesting, but it also is a little bit orthogonal to the remainder of the paper; at least with the current framing. The paper would be stronger if it could provide a more cohesive narrative in terms of incoporating this section.
- The use of BERT-base scale models naturally leaves open the question of whether anything changes at larger scale. While the fine-tuning paradigm becomes effective at these scales, recent research suggests that zero-shot capabilities tend to emerge at larger model sizes. Given that zero-shot capabilities are naturally connected to investigating the role of knowledge transfer in pre-training, there is a natural unresolved question of whether self-pretraining would fall behind if the experimental setup is upscaled by at least an order of magnitude. This is, of course, understandably hard to assess without enormous compute expenditures.

**Summary Of The Paper:**

The paper studies the effectiveness of self-pretraining, meaning applying pretraining *objectives* like masked language modeling to *datasets* specific for a single task. Remarkably, self-pretraining often achieves performance comparable to the conventional approach of pre-training on generalist web-scale text. This casts doubt on the proposition that knowledge transfer from large text corpora is at the core of why pre-training is so effective in NLP.

Contributions:
- The paper compares self-pretraining to using an off-the-shelf model like ELECTRA, and finds that the two often achieve comparable results.
- Pre-training on other task-targeted datasets (rather than generalist web corpora) is also shown to provide significant gains
- Self-pretrained models are observed to have a different distribution of errors compared to off-the-shelf pretrained models, but simple ensembling is not able to take advantage of this to get better results

**Summary Of The Review:**

Overall the paper contains a thorough and comprehensive experimental evaluation of self-pretraining, and the result that self-pretraining can be as effective as off-the-shelf models can have important implications for our understanding of why the pre-training paradigm is as effective as it is.

---

> ### Author Response · Authors · 2022-11-16
> **Response to Reviewer 8VQm**
>
> Thank you for your positive assessment and for championing our work. We are glad that you find our experimental evaluation comprehensive and the implications of our findings to be important.
>
> > **“The discussion on how to construct artificial long-term dependencies is very interesting, but it also is a little bit orthogonal to the remainder of the paper; at least with the current framing. The paper would be stronger if it could provide a more cohesive narrative in terms of incorporating this section.”**
>
> Thanks for your feedback. We have improved the exposition of the paper to accommodate your feedback. First, we would like to mention that the section on artificial long-term dependencies has been relegated to the Appendix to make room for extensive new results on structured prediction and commonsense inference tasks. Back to the reviewer’s concern, the main connection between constructing artificial long-term dependencies and the remainder of the paper is that it broadens our exploration of ways to use  the downstream datasets.
>
>
> > **“The use of BERT-base scale models naturally leaves open the question of whether anything changes at larger scale. … recent research suggests that zero-shot capabilities tend to emerge at larger model sizes … there is a natural unresolved question of whether self-pretraining would fall behind if the experimental setup is upscaled by at least an order of magnitude. This is, of course, understandably hard to assess without enormous compute expenditures.”**
>
> Thank you for your astute comment. We believe that this is an interesting suggestion for future investigation.
>
> > **The ensembling results appear to have ensembling take place over probabilities, rather than discrete predictions ... Is there any way to apply majority-vote style ensembling instead, to get around the issues with uncalibrated model confidence levels?**
>
> Yes, we are ensembling two models with their prediction probabilities after calibrating models with temperature scaling. Majority voting is an interesting suggestion. However, as you mentioned, when using the ensembles with just two models,  disagreeing predictions would lead to ambiguous output and we are not aware of any relevant technique that can handle ambiguous predictions without just arbitrarily breaking ties.

---

> ### Comment · Reviewer_8VQm · 2022-12-05
> **Response to authors**
>
> Thank you for your responses.
>
> Having looked over all of the other reviews, the author responses, and the new results presented in the updated paper, I'm inclined to keep my score.
>
> I think this paper has a valuable contribution to our understanding of how pre-training actually functions to achieve the progress that we've seen in the field. Naively, one might think that scale (in terms of dataset and model size) is all that matters for model effectiveness, and that the pre-training objective is just a clever hack to get around the fact that labels are not available for large quantities of raw text. What this paper demonstrates, especially with the addition of the "WikiSub" results, is that the pre-training objective is far from a hack but is in fact fundamental to the success of the current training paradigm. To put it another way: even in a low-resource language where every last bit of raw text in that language has been assigned a task-specific label, one would *still* apply self-pretraining rather than training a model from scratch.
>
> This is the kind of claim that I believe should be known to the community, and can have impact in guiding the direction of future progress in the field.

---

### Author Response · Authors · 2022-11-16
**Common Response**

We would like to thank all four reviewers for their constructive comments and suggestions. We were pleased to see that the reviewers found the theme of the paper to be interesting and important (R1,R3,R4), and that all reviewers noted the thoroughness of our empirical experiments across multiple datasets and models.
In particular, we were happy to see the reviewers note that our results hold across a variety of text domains and dataset sizes (R1,R3), and both ELECTRA and Roberta models (R2,R4). We were also pleased to see that the reviewers appreciated the usefulness of our work both for understanding the effectiveness of pre-training (R1) as well as mitigating concerns about training on huge corpora which potentially contain biases and toxic language (R4).

We are also grateful to the reviewers for their many constructive suggestions, which have helped us to make significant improvements to the manuscript. While address most of the reviewer’s concerns in the respective threads, we would like to briefly address some  of the common questions raised by the reviewers below:

> **Using more complex tasks than classification (R3)**

In Section 7 of the updated manuscript, we have added experiments for 4 new datasets,including evaluations on the following structured prediction and commonsense inference tasks:

1. SQUAD (Question answering)

2. CONLL-2012 (Named entity recognition)

3. SWAG (situated commonsense inference)

4. HellaSwag (situated commonsense inference)

We have added the results for Electra-small and Roberta-base models in Table 6. We find that for Electra-small, self-pretraining achieves greater than 70% of the (pretraining) benefit achieved by an off-the-shelf model on all datasets. For Roberta-base, self-pretraining achieves 40-80% of the benefit of a model trained on large-scale upstream corpus, depending on the dataset. These results show that a significant portion of pretraining’s benefits can be realized in the absence of transfer (from an upstream dataset). Notably, the gap between self-pretraining and off-the-shelf performance also leaves a significant role for transfer to play.

> **About the role of pretraining data scale and  pretraining on subsampled upstream corpus. (R2, R3, R4)**

Our primary argument is not about the size of pretraining corpus but about its source. Concretely, our most important contribution is to show that pretraining can be surprisingly effective, even when no additional data, beyond that already available for training the downstream predictive model is incorporated. As a secondary finding, we do examine how each of these task-specific pretrained models perform on the other tasks. While the sizes of the various datasets do vary, the interesting thing about our findings here is that size does little to explain which cross-task transfers will be effective. We would like to note that prior work has already given a thorough treatment to the question of how upstream corpus size influences the performance of fine-tuned models on downstream tasks (see Zhang et al. “When Do You Need Billions of Words of Pretraining Data?”. ACL 2021).

That said, we agree with reviewers that more can be done to show how these results might compare against pretraining on a size-adjusted general upstream corpus. Following reviewers’ suggestions, we updated our manuscript to include the following additional experiments:  for each downstream dataset, we pretrained an ELECTRA model using a random subset of Wikipedia equal in size to the downstream training set. We then fine-tuned that pretrained model on the downstream dataset. We have added the resulting performance numbers as a column in Table 2. In 8 cases, self-pretraining outperformed pretraining on a size-matched upstream corpus and in 2 cases, it performed worse.

---

> ### Author Response · Authors · 2022-11-16
> **Common Response (continued)**
>
> > **Concerns about whether we are claiming that knowledge transfer has no role to play in pretraining’s success / whether “linguistic knowledge” plays a role in pretraining’s success.(R2, R4)**
>
> We want to be very clear at the outset that we are not claiming that genuine “transfer” has no role to play in pretraining’s success. Rather, our exciting finding is that a significant portion (but not all) of pretraining’s benefits can be realized even in the absence of additional data.
>  To stress the point further, transfer clearly plays a role in many of the exciting new developments in NLP. For instance, the zero-shot model performance of large language models such as GPT-3 can only be due to transfer. Our more subtle claim is that pretraining has significant benefits that cannot be explained by cross-dataset transfer. Even when we pretrain using only the exact same data on which finetuning will be performed, we realize (sometimes transformative) benefits in  downstream performance compared to a non-pretrained model. Moreover, for several classification tasks, such self-pretrained models perform even better than off-the-shelf models trained on a huge amount of external data (Bookwiki corpus).
>
> In this work, we do not attempt to characterize the features learnt by the models during pretraining, i.e., we do not check whether they are learning linguistic or other kinds of knowledge. It is a separate line of research which many recent papers have debated (e.g. Maudslay et al 2021, Niu et al 2022). When we say that “knowledge transfer” is not necessary in many cases, we mean whatever it is  that makes pretrained models perform better on the downstream task,  need not be “transferred” from vast upstream corpora and can be simply learnt from the downstream training set itself using the pretraining objective. We have edited our draft to clarify this point.
>
> Maudslay et al. “Do Syntactic Probes Probe Syntax? Experiments with Jabberwocky Probing”, NAACL21
>
> Niu et al. “Does BERT Rediscover a Classical NLP Pipeline?”, COLING22
>
> > **Relevance of our results in the era of much larger models (R1, R2)**
>
> Our results show the efficacy of self-pretraining for Electra-small and Roberta-base models when compared to off-the-shelf models trained on the Bookwiki corpus. We can not say for certain how self-pretraining with perform relative to standard pretraining as model size and corpus size are scaled up. Due to limited computational resources, we are unable to do the experiments involving larger scales. We hope that our findings might inspire corporate research labs with more ample computational resources to investigate how these findings scale.
> We note that results at the current scale  hold both conceptual relevance for understanding the mechanism by which pretraining works, and practical relevance in domains (e.g., mobile deployment) where enormous models are not feasible.

---

### Decision · Program_Chairs · 2023-01-20

**Decision:**

Reject

**Justification For Why Not Higher Score:**

* The takeaway message from the paper is not that useful
* There are some doubts on how the performance can scale when larger scale of data is in picture

**Justification For Why Not Lower Score:**

n/a

**Metareview: Summary, Strengths And Weaknesses:**

The paper presents a set of experiments to show that when the finetuning dataset is also used for pre-training there are substantial gains that can be made in the performance on a give task. Of course, when the finetuning dataset is being used for pretraining it is being used without any labels, and only the MLM objective is being used for pretraining. The gains in performance are impressive and they come at no extra data. A new set of experiments that the authors added after the review show that if a similar size of randomly selected dataset is used for pretraining, most of the performance gap be covered.

Strengths
* clearly written paper
* thorough set of experiments
* good empirical performance gains

Weakenesses
* Similar gains in performance were also shown by Yao et al
* This could be more about the size of the dataset rather than the data being the same finetuning data.
* What happens if the size of the pretraining data is substantially larger? can these gains diminish

**Summary Of Ac-Reviewer Meeting:**

The reviewers acknowledged the efforts put in by the reviewers in experimentation, but there was a consensus that the results might not be very useful for others as these results assume absence of using an LLM as the baseline. Further the contribution of the paper did not seem to match the standard for ICLR without significant change in the story and framing of the paper.